# Improving provider and client communication around family planning in Togo: Results from a cross-sectional survey

Nicole Bellow [1]*, Leanne Dougherty[2], Dela Nai[3], Sethson Kassegne[4], Robert Hugues Yaovi Nagbe[4], Lorimpo Babogou[4], Kossi Mawuko Guede[5], Martha Silva[6]

1 Avenir Health, Takoma Park, Maryland, United States of America, 2 Population Council, Washington, DC, United States of America, 3 Population Council, Accra, Ghana, 4 CERA Group, Lomé, Togo, 5 Pathfinder International, Lomé, Togo, 6 Department of International Health and Sustainable Development, Tulane University, New Orleans, Louisiana, United States of America

* nbellows@avenirhealth.org

## Abstract

Previous research has shown that clients are better able to achieve their reproductive intentions when family planning (FP) services meet their needs and they have satisfying client provider interactions. There are several areas of quality provider-client communication, including providers taking a complete reproductive history of their clients to best gauge their needs, communication around alternative FP methods and side effects captured in the method information index, and communication around sexually transmitted infections and HIV risk as it relates to FP choices. This study examines data from a clinic-based intervention in Togo that focuses on strengthening health provider counseling related to FP, including improving in these three areas of provider-client communication. A clustered sampling approach was used to select 650 FP clients from 23 intervention facilities and 235 clients from 17 control facilities in the Lomé and Kara districts of Togo. The FP clients' interactions with providers were observed and clients exit interviews were conducted in December 2021. For each communication area measured through client interviews and observations, principal components analysis and Cronbach's alpha scores were used to ensure that the individual components could be indexed. Outcomes variables based on an index of sub-questions were then created for those who had fulfilled each of the components within an index. Multivariate multilevel mixed-effects logit models accounted for clients nested within facilities and included independent variables capturing client demographic and facility variables. Multivariate results show that all three outcome variables representing the three provider-client communication areas were statistically significantly better for FP clients in intervention clinics versus control clinics (p<0.05). The results speak to the emphasis that the Togo Ministry of Health has placed on building the provider capacity to provide quality counseling and administration of FP methods and working to assist in achieving health programming goals through well-designed interventions.

**Data Availability Statement:** All relevant data for this study is publicly available from the Harvard

Dataverse repository (https://doi.org/10.7910/DVN/EUQ6Y0).

**Funding:** The United States Agency for International Development (USAID) under the terms of the Breakthrough RESEARCH project cooperative agreement (AID-OAA-A-17-00018) provided funding for the research conducted for this manuscript and USAID provided a technical review of the draft of the manuscript. The following researchers work on activities funded under the Breakthrough RESEARCH project: NB, LD, SK, HRN, LB and MS. The AmplifyPF project is funded by a separate USAID cooperative agreement 72062418RFA00005 and provides funding for the following researchers: DN, KMG. The contents are the responsibility of the authors and do not necessarily reflect the views of USAID or the United States Government.

**Competing interests:** The authors have declared that no competing interests exist.

## Introduction

High maternal morbidity and mortality rates continue to burden Francophone West Africa, a sub-region characterized by having the highest fertility rates in the world with low contraceptive prevalence. In Togo, the total modern contraceptive prevalence rate among all women of reproductive age is estimated at 21% for 2020 [1]. The most recent estimates of unmet need for modern contraception in Togo are 33.0% for married women and 25.2% among all women of reproductive age [2].

Previous research has shown that clients are better able to achieve their reproductive intentions when family planning (FP) services meet their needs and they have satisfying provider-client interactions [3,4]. There are several areas that comprise high-quality provider-client communication. One communication area of technical quality is whether providers take a complete reproductive history of their clients to best gauge their needs [5,6]. Another increasingly used measure of quality provider-client communication is the validated three item method information index (MII) or the four item MIIplus, which assess the extent to which information about FP methods is communicated to clients [7,8]. The MII and MIIplus have been found to be positively associated with better FP continuation in Pakistan and India and the MII has also been found to be positively associated with voluntary use of long-acting reversible contraception (LARC) [7,9–12].

An additional aspect of FP provider-client communication is assessing the client's HIV/STI disease risk, including inquiring about the partner's status [13,14]. While many have advocated for the integration of HIV/STI risk assessment into FP counseling and promotion of "dual protection" via condoms and other forms of FP, several studies acknowledge that who presents at the facility, and their reason for going to a facility, may influence how to best integrate services [15–17].

Programs aiming to improve FP provider-client communication can take several approaches. In Togo, and three other countries, the United States Agency for International Development (USAID)-funded AmplifyPF project is implementing the Integrated Learning Network (ILN) model, which is based on a multisectoral approach that convenes stakeholders at the district level to coordinate resources and ensure the delivery of high-quality FP services. In Togo, the ILN model is being implemented in five health districts across the country (intervention sites) and includes building health provider capacity through training on FP counseling, and delivery. Using tools developed by the Togo Ministry of Health (MoH) and the project itself, training activities entailed strengthening health provider counseling related to FP through counseling models that focus on enabling the provider to establish rapport with the client, exploring the needs of the client, and providing support for decision making and application of the decision. In identifying the needs of clients, the FP providers are trained to complete a comprehensive client history, assess client risk for HIV/STO, and to cover the 4 components of the MII+ which comprise providing counseling about FP methods including dual protection, potential side effects and what to do if they experience side effects, including options to switch methods [18].

This study examines the impact of AmplifyPF's targeted training on provider-client communication in health facilities across its intervention sites in the Lomé districts of Togo. The primary hypothesis is that clients seeking FP services in clinics that received the AmplifyPF intervention will experience higher quality communication from providers than clients seeking FP services in clinics that did not receive the intervention.

## Materials and methods

Data for this analysis come from the quantitative component of a mixed-methods cross sectional study designed to compare intervention and control sites on the quality of

communication between providers and clients. The Ministry of Hygiene and Public Health and Universal Access to Care in Togo provided approval for the study and consent forms (No. 033/2021/CBRS). The study also received approval from the Population Council Institutional Review Board in the United States (No. 985).

A pre-test power calculation was originally conducted to detect a 14% difference in client satisfaction that resulted in an estimated sample size of approximately 600 female FP clients in the intervention area and approximately 400 clients in the comparison area (total N = 1,000) per round. When the client satisfaction variables did not yield any useful findings due to nearly universal reporting of satisfaction with their FP visit, we conducted a post-test power calculation and determined the sample size detected a difference of 10% points in the proportion of provider communication topics observed in the intervention area, with 80% power to detect a difference, alpha of 0.05 and assuming a design effect of 1.2 to account for intracluster correlation [19]. Based on these results, we estimated a sample size of approximately 600 female FP clients in the intervention area and 400 clients in the comparison area for a total of N = 1,000 client-provider observations and client exit interviews [20]. Both client observations and exit interviews were used to capture process attributes around the delivery of care as well as client satisfaction.

A clustered sampling approach was used to randomly select 23 of 46 intervention facilities in three of the five AmplifyPF ILN districts. Data collection at intervention facilities took place on "FP Special Days" when facilities experienced higher client volumes because contraception was provided for free. A convenience sample of approximately 28 clients were selected per intervention facility, resulting in a final sample size of 650 female FP clients from the intervention area. Seventeen comparison facilities were selected to match characteristics to the selected intervention facilities, including facility type, population size served, level of urbanization, client volume, and staff size. Due to lower than anticipated client volume in the comparison area, a final convenience sample resulted in 235 female FP clients in the control areas with approximately 13 clients per facility.

Beginning in late 2019, AmplifyPF conducted monitoring and supervision of FP provider counseling across health facilities. As part of its continuous quality improvement approach, formal training on counseling at facilities began in September 2020 and continued until the end of 2021. At the facilities in districts where AmplifyPF operates and control facilities in non-AmplifyPF districts, FP clients were enrolled for client observations and exit interviews in December 2021, conducted by trained interviewers. To facilitate enrolling clients, receptionists informed clients that they may be asked to participate in an anonymous observation and survey during their visit, but it was not compulsory, and it would not affect their services received. If the client agreed, before each participant was observed (client and provider) and interviewed, an informed consent statement was read by the participant (or the interviewer if the participant is illiterate) and signed by both participants (provider and client). The interviewers conducted provider-client observations to capture different topics of provider-client communication, including questions asked by the provider to assess the client's reproductive history and provider-client communication around sexual partners and HIV/STI risk. The survey instruments were adapted from global standard questions and sequences employed by the Demographic and Health Survey Service Provision Assessment (SPA). Client exit interviews assessed client-reported information given to them by providers about FP method side effects and alternative methods, as captured in the Method Information Index Plus (MIIplus). Both the client observations and exit interviews assessed aspects of client privacy and confidentiality, with privacy referring to structural privacy in terms of ensuring visual and auditory privacy.

The observations and exit interviews were matched by client and combined into a single client-level dataset. Within each communication topic measured, principal components analysis and Cronbach's alpha scores were used to ensure that the individual components could be combined into one index outcome variable. Each index outcome was constructed by assigning a 0 (no) or 1 (yes) to each component question and summing across all questions. Binary index outcome variables were then created for those who had responded yes (1) to each of the component questions within an index. The indices were then used to compare provider-client communication among clients at intervention and control sites. Binary outcome variables were generated, as opposed to ordinal scales, for ease of interpretation. The three final outcome variables examine provider-client communication directly addressed in the AmplifyPF training, including:

- *Providers taking a complete patient history relevant to FP*–eight components: the last delivery date, last menstrual period, breastfeeding status, regularity of menstrual cycle, age of client, number of living children, desire for a child or more children, and desired timing for birth of next child, as adapted from the Demographic Health Survey service provision assessment survey [21].

- *Partner's status and risk of STIs/HIV*–four components: the partner's status in terms of number of partners and periods of absence from partner, client's perceived risk of STIs/HIV, use of condoms to prevent STIs/HIV, and using condoms along with another method to prevent both pregnancy and STIs/HIV.

- *MIIplus*–four components: the provision of information about different FP methods, the possible side effects/problems with the selected method, what to do if one experiences side effects, and discussing the possibility of switching to another method if the selected method is not suitable [22].

Data analysis was conducted using STATA 16.1. Bivariate results controlled for the clustered nature of the data. For multivariate results, multilevel mixed-effects logit models were used to account for clients nested within facilities and explanatory variables were added for age, education, FP status at start of visit (not using FP, considering switching FP method, continuing current method), privacy and confidentiality, and facility staff size.

## Results

In total, 650 client observations and exit interviews were conducted in the intervention facilities and 235 were conducted at the control facilities. At the facility-level, clients were served primarily by midwives or nurses with on average 10 years of experience. **Table 1** details the client-level variables by intervention and control facilities. Overall, the results show comparable client characteristics, albeit a somewhat greater proportion of clients under age 25 being served in intervention areas (27%) versus control areas (22%). There were also statistically significant differences in the education status of clients, with a greater proportion of clients having no education (13%) and university education (13%) in intervention facilities compared to the control facilities.

The bivariate results in **Table 2** compare the individual components of the three outcome variables between clients at the intervention facilities and clients at the control facilities. With regards to a provider taking a complete history, there was a substantial variation in the overall sample on the proportion of providers who asked key questions about the client history. The most frequently asked question was the date of the last menstrual period (83%); however, far fewer clients were asked about current breastfeeding status (44%) or the desired timing for

**Table 1. Descriptive client and facility variables.**

| | Overall (N = 885) | Intervention (N = 650) | Control (N = 235) | p-value |
|---|---|---|---|---|
| *Client variables* | % | % | % | |
| Age of client [c] | | | | |
| Under 25 years | 25.9 | 27.4 | 21.7 | 0.032 |
| 25–34 years | 48.3 | 48.9 | 46.4 | |
| 35+ years | 25.9 | 23.7 | 31.9 | |
| Education (highest level attended) [c] | | | | |
| None | 11.8 | 12.8 | 8.9 | 0.010 |
| Primary school | 26.4 | 24.6 | 31.5 | |
| Secondary school | 50.5 | 49.7 | 52.8 | |
| Superior/university | 11.3 | 12.9 | 6.8 | |
| Provider ensured privacy and confidentiality [c,o] | | | | |
| Yes | 75.4 | 75.1 | 76.2 | 0.739 |
| No | 24.6 | 24.9 | 23.8 | |
| Client FP status at visit outset | | | | |
| Not using FP | 26.6 | 25.4 | 29.8 | 0.111 |
| Using FP, considering switching methods | 19.7 | 21.2 | 15.3 | |
| Using FP, not considering switching methods | 53.8 | 53.4 | 54.9 | |

c = client survey; o = observation data.

one's next birth (53%). For nearly all the components to client history, clients in intervention sites were asked the individual questions at a greater rate than clients in the control facilities, except for providers asking about the last menstrual period. Approximately 32% of all clients

**Table 2. Provider-client communication.**

| | Overall (N = 885) | Intervention (N = 650) | Control (N = 235) | p-value |
|---|---|---|---|---|
| *Client history [o]–provider asked client about:* | % | % | % | |
| Last delivery date or age of youngest child | 72.7 | 82.8 | 44.7 | <0.001 |
| Last menstrual period | 83.2 | 84.2 | 90.4 | 0.191 |
| Breastfeeding status | 44.1 | 46.8 | 36.6 | 0.007 |
| Regularity of menstrual cycle | 60.3 | 62.5 | 54.5 | 0.032 |
| Age of client | 74.8 | 79.1 | 63.0 | <0.001 |
| Number of living children | 64.4 | 71.4 | 45.1 | <0.001 |
| Desire for a child or more children | 62.0 | 69.7 | 40.9 | <0.001 |
| Desired timing for birth of next child | 53.1 | 62.3 | 27.7 | <0.001 |
| Complete history (all questions) | 31.6 | 36.3 | 18.7 | <0.001 |
| *Method information index plus [c]–provider:* | | | | |
| Provided information about different FP methods | 72.1 | 79.2 | 53.5 | <0.001 |
| Talked about possible side effects or problems with the selected method | 73.6 | 81.1 | 62.2 | <0.001 |
| Told you what to do if you experienced any side effects of problems with the selected method | 77.5 | 83.2 | 62.2 | <0.001 |
| Talked about the possibility of switching to another method if the selected method was not suitable | 74.2 | 79.0 | 61.8 | <0.001 |
| Method information index plus (all questions above) | 55.5 | 62.3 | 36.6 | <0.001 |
| *Partner status and HIV/STI risk [o]–provider asked client about:* | | | | |
| Partner HIV/STI status | 35.9 | 40.3 | 23.8 | <0.001 |
| Client's perceived risk of HIV/STIs | 34.6 | 40.5 | 18.3 | <0.001 |
| Use of condoms to prevent HIV/STIs | 27.9 | 32.8 | 14.5 | <0.001 |
| Using condoms along with another method to prevent both pregnancy and HIV/STIs | 22.0 | 26.3 | 10.2 | <0.001 |
| All questions above | 16.6 | 20.2 | 6.8 | <0.001 |

**Table 3. Multivariate logistic regression results with adjusted odds ratios (N = 885).**

| Variable | Outcome 1: Complete FP history | Outcome 2: MIIplus | Outcome 3: HIV/STIH risk |
|---|---|---|---|
| Intervention | | | |
| Intervention facility client | 4.93* | 3.93* | 7.25* |
| Control facility client | — | — | — |
| Client age | | | |
| Under 25 years | — | — | — |
| 25–34 years | 0.89 | 1.39 | 0.65 |
| 35 plus years | 0.65 | 1.05 | 0.84 |
| Client education | | | |
| Primary or less | 1.21 | 0.82 | 0.81 |
| Secondary or more | — | — | — |
| Ensured privacy and confidentiality | | | |
| Yes | 1.38 | 3.52*** | 3.77** |
| No | — | — | — |
| FP status at start of visit | | | |
| Not using FP | 3.23*** | 6.02*** | 2.36*** |
| Using FP, considering switching methods | 2.67*** | 7.25*** | 2.93*** |
| Using FP, not considering switching methods | — | — | — |

Note–models control for facility staff size. * p<0.05, **p<0.01, ***p<0.001.

were asked all the relevant questions, with 36% among clients in intervention facilities and 19% of those in control facilities. When examined together, the client history questions had a Cronbach's alpha score of 0.89 and all components loaded onto into one factor when conducting principal components analysis.

Similarly, the individual components for MIIplus index outcome questions as well as partner status and HIV/STI risk all showed greater provider-client communication among clients in the intervention facilities versus the control facilities. While over 56% of the overall sample reported communication about all four of the MIIplus components (62% in intervention and 37% in control), far fewer were asked comprehensive questions about partner status and HIV/STI risk. In total just 17% of the overall sample reported being asked all four questions, 20% for those in intervention facilities and 7% in control facilities. For both the four-component MIIplus variables and partner status/risk index variables, the sub-component loaded into one factor for each category and had a Cronbach's alpha score of 0.85 and 0.87, respectively.

The multivariate logistic regression results for each of the three outcome variables are shown in **Table 3** (see S1 File for more details). For all three outcome variables, attending an intervention facility increased the odds of having every component of the index addressed and this association was statically significant (p<0.05). For outcome 1, clients in intervention facilities had nearly five times the odds of having a complete FP history compared to those in control facilities (OR = 4.93). One's FP status at the beginning of the visit was also statistically significant for this outcome. Clients not currently taking FP at the beginning of the visit and clients considering switching FP methods both had higher odds for the outcome variable, with ORs of 3.23 and 2.67, respectively (p<0.001).

For outcome 2, the results show that those in intervention facilities had nearly four times the odds of having been asked all four of the MIIplus questions (OR = 3.93). Other statistically significant variables include that the client's privacy and confidentiality was ensured (OR = 3.52), the client was not using FP at the beginning of the visit (OR = 6.02), and that the client was considering switching methods (OR = 7.25).

Finally, for outcome 3, clients in intervention clinics had seven times the odds of being asked all the partner and HIV/STI risk questions compared to control clients (OR = 7.25). As

with outcome 2, the other statistically significant variables include that the client's privacy and confidentiality (OR = 3.77), not using FP at the beginning of the visit (OR = 2.36) and considering switching methods (OR = 2.93).

## Discussion

When looking at individual communication components, certain areas of communication are discussed between providers and clients more than others. Communication was strongest for the MIIplus question related to FP options and side effects, with more than 70% of clients reporting communication around each of the four MIIplus components on their own. The results reflected in the MIIplus also indicate that the most comprehensive communication is taking place around FP methods, use, and side effects, with over half of clients being asked all four questions. Given that the MIIplus measure has been validated to reflect high quality FP counseling [7,9] these findings bode well for the intervention clinics that show stronger communication by this metric. The variation in the MII and MIIplus metrics and their association with the AmplifyPF intervention is particularly useful given that the client satisfaction measures were uniformly high and thus not suitable for data analysis. The client history questions also have fairly high levels of asking about the last menstrual period and delivery, but also show that there is substantial room for improvement in inquiring about breastfeeding status and desired timing for next pregnancy.

As shown in the prior literature on provider-client communication around reproductive health care, the weakest area for communication at present is around the risks for HIV and STIs, particularly as it relates to HIV/STI risk and the use of condoms. Although Togo has a relatively low HIV prevalence rate at 2.7% for women of reproductive age, the last DHS showed that 13.8% of women reported having an STI or STI symptoms [21]. As such, it is critical that providers communicate to FP users about their HIV/STI risk and the options for using condoms for protection [21,22]. Given the connection between STIs and infertility, incorporating communication about HIV/STI risk and protection is an important part of comprehensive reproductive health that allows women and men to achieve their reproductive goals [23,24].

There are three key findings from the multivariate analysis. The primary result of this analysis is that clients receiving FP services in intervention clinics were statistically significantly more likely to have high quality provider-client communication, as measured by the three outcome variables. As such, it appears that the intervention was successful at improving key aspects of provider-client communication. Prior to the intervention, AmplifyPF worked closely with the MoH to identify gaps in provider capacity and together reviewed, revised, or developed relevant tools to implement as part of the intervention. AmplifyPF also conducts systematic supervision of providers and is often accompanied by members of the district health management team. Based on qualitative feedback collected from providers in the target ILNs, they have generally welcomed the interrelated elements of the intervention, particularly the capacity building and supervision as well as site walkthroughs conducted by community stakeholders that highlight quality competencies. As such, providers in the intervention sites not only recognize the impact of improved provider-client communication but also that they will be held accountable during stakeholder engagement activities. The confluence of all these elements are likely motivations for the providers.

In addition to the intervention itself, the second key finding is that provider-client communication is enhanced when privacy and confidentiality are ensured. This finding is consistent with prior literature on the importance of structural components of quality of care and demonstrates that, in addition to provider training, structural improvements that ensure privacy and confidentiality can also be critical to enhance the quality of FP counseling [25,26].

Third, the results show that providers provide more thorough provider-client communication to clients who are not currently taking FP and those considering switching methods receive more thorough communication based on these outcome variables. This result is not surprising in that a client starting a new FP method, whether due to method switching or a new FP user, will require more information than a client continuing a method. As such, some providers likely triage their limited time accordingly. Still, it is important that providers strive for thorough communication for all clients by asking questions, as some clients may be experiencing difficulties with their current FP method and would be better served by a different method but are reluctant to bring it up with a busy provider.

As with all research, there are limitations that must be acknowledged. First, the results shown here are cross-sectional and therefore one cannot assume causality between the intervention and the communication outcomes. Additionally, a few intended control variables were inadvertently left off the surveys and therefore the models were unable to control for clients' marital status, and parity. We also did not control for client volume, as facilities sampled reported similar client volume prior to the study; however, during data collection, the estimated client volume in the control facilities was significantly lower than planned for and as a result led to a smaller sample size of N = 235 compared to the planned sample size of N = 400. Furthermore, there is potential bias from client observations when providers know they are being observed, as with this study, although this bias would be present in both the intervention and control facilities. Client exit interviews are also subject to recall bias, even in situations such as this study where the exit interview occurred immediately after the service.

## Conclusions

Despite these limitations, the data clearly demonstrate a greater odds of a client receiving high quality communication, as measured by three outcome variables reflecting information exchange through a complete FP history, discussions about FP methods, use, and side effects, and discussions about clients' partners and HIV/STI risk status, in the intervention clinics compared to the controls. These results do not include the totality of high-quality service provision, but they do speak to the emphasis that the MoH has placed on building the provider capacity to provide quality counseling and administration of FP. The significant differences between intervention and control facilities shown here demonstrate that interventions focused on improving provider-client communication can yield positive results, which should be shared with stakeholders and beneficiaries alike through channels available to them, including community dialogues. Future work should be done to link improvements in communication to greater client satisfaction with FP services, increased uptake and continuation of FP, and for the ability of women to maintain their aspirations with regards to family size and spacing.

## Supporting information

**S1 File. Full regression output.**
(DOCX)

## Acknowledgments

We would like to thank additional members of the research team at CERA Group for their data collection efforts that enabled this study to be completed.

## Author Contributions

**Conceptualization:** Nicole Bellow, Leanne Dougherty, Martha Silva.

**Data curation:** Sethson Kassegne.

**Formal analysis:** Nicole Bellow.

**Funding acquisition:** Martha Silva.

**Investigation:** Dela Nai, Sethson Kassegne, Robert Hugues Yaovi Nagbe, Lorimpo Babogou, Kossi Mawuko Guede.

**Methodology:** Nicole Bellow, Leanne Dougherty, Dela Nai.

**Project administration:** Sethson Kassegne, Robert Hugues Yaovi Nagbe, Lorimpo Babogou.

**Resources:** Kossi Mawuko Guede.

**Supervision:** Leanne Dougherty, Dela Nai, Robert Hugues Yaovi Nagbe.

**Validation:** Kossi Mawuko Guede.

**Writing – original draft:** Nicole Bellow, Leanne Dougherty, Dela Nai.

**Writing – review & editing:** Nicole Bellow, Leanne Dougherty, Dela Nai, Sethson Kassegne, Robert Hugues Yaovi Nagbe, Lorimpo Babogou, Kossi Mawuko Guede, Martha Silva.

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
