## [Decision Letter · Decision Letter 0]

2 Feb 2023

PGPH-D-22-01744

Improving provider and client communication around family planning: Results from an intervention in Togo

Dear Dr. Nicole Bellows,

Thank you for submitting your manuscript to PLOS Global Public Health. After careful consideration, we feel that it has merit but does not fully meet PLOS Global Public Health’s publication criteria as it currently stands. Therefore, we invite you to submit a revised version of the manuscript that addresses the points raised during the review process.

Please, do find in this e-mail the reviewers comments, questions and suggestions to be made to the manuscript in order to bring more clarity to the arguments and the presentation of the research results, the ethics approval and methodology. 

Both reviewers and myself agree that it is a well written manuscript, but please do make sure to review again, for example, there are minor concerns relating to the use and description of acronyms first time used in the manuscript.

This is a seminal works that goes in line with the criteria and principles followed by PLOS Global Public Health journal. 

We look forward to receiving your revised manuscript.

Kind regards,

María De Jesús Medina Arellano, PhD

Academic Editor

Journal Requirements:

2. We have noticed that you have uploaded Supporting Information files, but you have not included a list of legends. Please add a full list of legends for your Supporting Information files after the references list. 

3. In the online submission form, you indicated that "The data underlying this analysis will be available on the USAID Development Data Library (https://data.usaid.gov/). The anonymized data are scheduled to be uploaded to the library by March 2023 but can be made available prior to this date upon request.". All PLOS journals now require all data underlying the findings described in their manuscript to be freely available to other researchers, either 1. In a public repository, 2. Within the manuscript itself, or 3. Uploaded as supplementary information.

Reviewers' comments:

Reviewer's Responses to Questions

**Comments to the Author**

1. Does this manuscript meet PLOS Global Public Health’s publication criteria? Is the manuscript technically sound, and do the data support the conclusions? The manuscript must describe methodologically and ethically rigorous research with conclusions that are appropriately drawn based on the data presented.

Reviewer #1: Yes

Reviewer #2: Yes

2. Has the statistical analysis been performed appropriately and rigorously?

Reviewer #1: Yes

Reviewer #2: Yes

3. Have the authors made all data underlying the findings in their manuscript fully available (please refer to the Data Availability Statement at the start of the manuscript PDF file)?

Reviewer #1: Yes

Reviewer #2: Yes

4. Is the manuscript presented in an intelligible fashion and written in standard English?

Reviewer #1: Yes

Reviewer #2: Yes

5. Review Comments to the Author

Reviewer #1: This is an interesting and well written manuscript. I do have some questions and suggestions for your consideration.

1. I see that you have conducted post-test power calculations. Did you also conduct pre-test power calculations? If not, is there a reason why not, and if so, is there a reason why you chose to present the post-test calculations instead? In that same vein, can you explain why the client volume in the control facilities was so much lower than expected.

2. Can you please elaborate on the intervention and how it relates to the outcomes you were measuring. Can you also please describe when the intervention occurred in relation to the survey? Was the survey immediately after the intervention, or had some time passed, which might suggest sustainability of results? Also, you state in the discussion that the providers have welcomed the interrelated elements of the intervention. Did you collect quantitative or qualitative feedback from providers to assess their satisfaction with the intervention?

3. Was there a reason why you conducted both observations of client visits and client exit interviews? If you were observing visits, could MII and MIIplus have been collected during those visits instead of at a separate CEI?

4. Depending on how it is defined, "ensuring privacy" could be either a structural quality component or a process quality component. I think that you are considering privacy to be a structural quality component, but it would be helpful to describe how you are defining privacy in this context.

5. You allude to this in the conclusion, but the ultimate goal of improved provider communication is improved client satisfaction, higher FP uptake/continuation, or both. Did you measure either of those outcomes? Are they correlated with the improved communication that you observed at intervention facilities?

Reviewer #2: General comments

The manuscript addresses an important and sometimes neglected issue namely early childhood caries. The findings of an in-depth scoping review focusing on the Gulf Cooperation Council (GCC) countries is resented. While the summary of the current knowledge reported is compelling, there are concerns in the way the way the evidence is presented.

Specific Comments

Title – The title could be improved to include the study design used to answer the research question – cross sectional survey (as described in the methods). The portion “Results from an intervention in Togo” is not very informative.

Abstract – The abstract is well written and summarizes the rationale for the study, methodology used, key findings, and conclusions. However, the key findings could be strengthened by including some of the significant results from the multivariate analysis i.e. ORs, p-values, etc.

Introduction - The authors have cited relevant studies and provided good background, focusing on the main focus of the manuscript.

Materials and Methods –

- The paragraph on ethical approval seems oddly phrased for a manuscript, e.g. remove language referencing the editorial office, among other things --- rephrase. “The relevant ethical approval and consent details were received and are available on request by the editor or editorial office. Study participants provided informed consent by using their signature. In addition, all methods were carried out in accordance with relevant guidelines and regulations and with the 1964 Helsinki declaration and its later amendments or comparable ethical standards.”

- Outcome variables and the analysis conducted are well defined.

Results – Summary table presented is well laid out and good accompanying text highlighting the key findings.

Discussion/ Conclusion – These two sections frame the findings well in answering the research question.

References – Relevant and recent.

Other comments – Minor editing required; authors to review the entire manuscript and make the relevant changes. For example, abbreviation MOH first appears on line 76, then it’s defined later in line 238. This should be the other way round.

6. PLOS authors have the option to publish the peer review history of their article (what does this mean?). If published, this will include your full peer review and any attached files.

**Do you want your identity to be public for this peer review?** For information about this choice, including consent withdrawal, please see our Privacy Policy.

Reviewer #1: No

Reviewer #2: No

---

## [Editor Report · Decision Letter 1]

10 May 2023

Improving provider and client communication around family planning in Togo: Results from a cross-sectional survey

PGPH-D-22-01744R1

Dear Dr. Bellows,

We are pleased to inform you that your manuscript 'Improving provider and client communication around family planning in Togo: Results from a cross-sectional survey' has been provisionally accepted for publication in PLOS Global Public Health.

Best regards,

María De Jesús Medina Arellano, PhD

Academic Editor